# Metabolizable Energy and Amino Acid Digestibility of Soybean Meal from Different Sources for Broiler Chickens Supplemented with Protease

**DOI:** 10.3390/ani14050782

**Published:** 2024-03-01

**Authors:** Maurílio de Lucas Xavier Junior, Rafael de Sousa Ferreira, Levy do Vale Teixeira, Jean Kaique Valentim, Kaique Moreira Gomes, Romário Duarte Bernandes, Arele Arlindo Calderano, Luiz Fernando Teixeira Albino

**Affiliations:** 1Department of Animal Science, Universidade Federal de Viçosa, Viçosa 36570-900, MG, Brazil; maurilio_junior1@hotmail.com (M.d.L.X.J.); kaique.tim@hotmail.com (J.K.V.); kaique.gomes@ufv.br (K.M.G.); duarteromario040@gmail.com (R.D.B.); calderano@ufv.br (A.A.C.); lalbino@ufv.br (L.F.T.A.); 2DSM Nutritional Products Brazil, Innovation and Applied Science, Mairinque 18120-000, SP, Brazil; levy.teixeira@dsm.com

**Keywords:** enzyme, nutrition, poultry, protease, supplementation

## Abstract

**Simple Summary:**

Due to the use of enzymes in the diet of broilers, antinutritional factors present in soybean meal, such as inhibitors of trypsin, can be reduced. Protease, for example, can reduce the inhibitors of trypsin in soybean meal and improve the performance of broilers. In this situation, this study aimed to evaluate the values of metabolizable energy (apparent and corrected for nitrogen balance) and amino acid digestibility of soybean meal samples from different regions, without and with the addition of protease in feeds for broilers with 14 to 28 days of age. The supplementation of soybean meal with protease resulted in higher values of metabolizable energy, both apparent and corrected by nitrogen balance. No interaction was observed between the factor’s soybean meal and enzyme for all of the variables presented. The protease supplementation improved (*p* < 0.05) the standardized ileal digestibility of essential amino acids compared to non-supplemented soybean meal, although the effect of protease varied among the amino acids. Protease supplementation in broiler diets may result in higher values of metabolizable energy and better digestibility of some essential amino acids.

**Abstract:**

This study investigated the effect of the serine protease on metabolizable energy and amino acids’ digestibility of different soybean meal for broilers. A total of 684 broilers chickens form 14 to 23 d age were distributed with nineteen treatments, six replicates, and six birds per replicate. Nine samples of soybean meal from different regions in Brazil were used, with some samples supplemented with the protease enzyme and others without addition. Apparent and corrected-for-nitrogen-balance metabolizable energy were evaluated, as well as the coefficients of amino acid digestibility. All collected data were submitted to ANOVA at a significance level of 5% and Tukey’s test was applied. The results showed that the addition of the protease enzyme significantly increased the values of AME and AMEn in all soybean meal samples. The soybean meal of different origins has significant variations in AME and AMEn. The addition of the protease improved the digestibility of essential amino acids compared to soybean meal without enzyme addition. These results indicate that supplementation with serine protease can improve the metabolizable energy and amino acid digestibility of soybean meal from different regions in the diet of broilers, potentially being an effective strategy to enhance nutrient utilization and animal performance.

## 1. Introduction

In poultry nutrition, soybean meal is one of the main protein sources used due to several factors such as the percentage of crude protein (CP), its good amino acid profile, and its bioavailability [1]. However, the existence of antinutritional factors (AF) in soybean meal has been shown to have a considerable effect in reducing the bioavailability of elements in this product, such as phosphorus [2]. There is a variety of AF, including lectin, raffinose, phytate, non-starchy polysaccharides, and trypsin inhibitors, among others [3]. Trypsin inhibitors, for example, impair protein digestion; their presence is characterized by hypertrophy of the pancreas due to the stimulation of pancreatic secretion and impaired growth of the animal [4].

The concentration of undigested proteins in the gastrointestinal tract requires nutritional management aimed at reducing damage to the animal’s digestive system, which can lead to losses in production [5]. The supplementation of exogenous proteases in broiler diets seems to be an excellent nutritional management method to avoid such losses [6]. Proteases are responsible for catalyzing the hydrolysis of peptide sequences and making them bioavailable [7]. Furthermore, improvements in apparent metabolizable energy (AME) levels are observed in diets containing exogenous proteases, confirming their ability to break protein–starch bonds in cereal grains [8]. In addition, these proteases have the potential to improve amino acid digestibility in broiler diets [9].

Therefore, this study hypothesizes that the use of protease results in the improvement of metabolizable energy, in addition to showing improvements in the amino acid digestibility of soybean meal. This study stands out by focusing on the role of proteases in various soybean meals, providing a specialized exploration of their effects on key nutritional parameters, thereby contributing innovative insights to the existing body of research in poultry nutrition. Thus, the objective was to evaluate the values of AME, apparent metabolizable energy corrected for nitrogen balance (AMEn), and amino acid digestibility of soybean meal samples from different regions, without and with the addition of protease in feeds for broilers.

## 2. Materials and Methods

### 2.1. Enzyme

The enzyme RONOZYME^®^ ProAct (200 g/ton of feed) (DSM Nutritional Products Brazil, Mairinque, SP, Brazil) was used on top. This protease is produced by the fermentation of *Bacillus licheniformis* containing genes transcribed from *Nocardiopsis prasina*. The enzymatic activity for this enzyme is defined by the amount of enzyme required to degrade 1 μmol of p-nitroaniline from 1 μM of the substrate (Suc-Ala-Ala-Pro-Phe-N-succinyl Ala-Ala-Pro-Phe-p-nitroanilide) per minute at a pH of 9.0 and temperature of 37 °C. The product used has 75,000 protease units/g of the enzyme.

### 2.2. Birds, Experimental Design and Diets

One-day old male Cobb-500^®^ broilers were raised until 14 days of age in a masonry house divided into protected circular pens containing a litter of wood shavings, feeding tubes, and manual drinkers with ad libitum access to feed and water. The lighting program consisted of 24 lights, and florescent lamps were used to provide artificial light at an intensity of 10-lux 24 h day. Temperature was maintained at 32 °C for the first week and then gradually lowered as recommended in the Cobb^®^ manual. The pre-starter feed consisted of corn and soybean meal and was in accordance with the nutrient recommendations of Rostagno et al. [10].

The amount of soybean meal used in the basal diet was replaced with one of the nine soybean meals used in this experiment

At 14 days, 684 birds were weighed (470 ± 80 g) and distributed in a completely randomized design using a factorial arrangement with 9 × 2 and 1 reference diet, a basal diet (Table 1), and nine samples of soybean meal from different regions of the country, totaling 19 treatments and 6 replicates of six birds per experimental unit (*n*: 36 broilers per group). Treatments were formulated by the addition of nine different soybean meals in the basal diet, replacing the amount of soybean meal used in the basal diet with one of the nine soybean meals used in this trial. Soybean meal was obtained from: Uberlândia; Rio verde; Toledo; Videira (Santa Rosa); Marau; Francisco Beltrão; Catanduvas; Chapecó; Videira (Olfar). Birds were housed in wire floor cages (500 cm^2^/bird) in a four-level battery equipped with a trough feeder and a nipple drinker for the collection of excreta and to determine AME, nitrogen-corrected apparent metabolizable energy (AMEn), and amino acid apparent and standardized digestibility coefficients. The experimental period extended from day 14 to day 28 of chick life.

### 2.3. Metabolizavel Energy Determination

From d14 to d23, the AME, nitrogen-corrected AMEn on dry matter, and AMEn on natural matter were determined. The feed was provided ad libitum for 10 days. For five days, the birds had an adaptation period of the feed and another five days for total excreta collection in each experimental unit twice daily, according to Sakomura and Rostagno [11]. Plastic-coated aluminum trays were placed under the cages to collect the excreta. The collected excreta were placed in plastic bags, labeled according to the experimental unit and stored in the freezer until the end of the collection period. Feed intake was measured during the period of the excreta collection.

Excrements collected in each cage were weighed at the end of the experimental period and homogenized for the energy test. To do so, 200-g samples were pre-dried at 55 °C for 72 h and ground in a ball mill (Tecnal Equipamentos para Laboratório, TE-350, São Paulo, Brazil) for 5 min until they turned into a fine mix.

Feed and excreta were analyzed to determine dry matter (DM) and CP [12]. The Kjeldahl method was used to determine the nitrogen content in both feed and excreta according to the official analysis methods [12]. The nitrogen excreted (NE) was calculated by multiplying the total amount excreted (in DM) by the percentage of nitrogen found in the excretion (also in DM). The same method was used to calculate the nitrogen intake (NI).

The nitrogen balance (NB) was based on the amount of nitrogen consumed and nitrogen excreted. The gross energy values (GE) were determined using a C5001 adiabatic calorimetric pump (IKA-Werke GmbH & Co., KG, Staufen, Germany). The values of AME and AMEn were calculated based on the GE values obtained for feed and excreta using the equations described by Sakomura and Rostagno [11]:AME = (GE_ing_ − GE_exc_)/DM_ing_ and
AMEn = (GE_ing_ − GE_exc_ − (8.22 × NB))/DM_ing_

In which GEing = gross energy ingested, GEexc = gross energy excreted, and DMing = dry matter ingested.

### 2.4. Amino Acid Digestibility Determination

The apparent and standardized digestibility coefficients of amino acids were determined from the 23rd to 28th days. Aiming to determine endogenous amino acid losses and determine the standardized amino acid digestibility, a protein-free diet (PFD) was formulated (Table 2). Soybean meals evaluated in this trial were incorporated into PFD, replacing 40% of the corn starch. In all experimental diets, Celite^TM^ insoluble acid ash was added at a level of 1%, according to Sakomura and Rostagno [11].

After the 4-day adaptation period, all birds used in the trial were slaughtered by electronarcosis for the collection of ileal digesta. For this, the birds were placed in the abdominal cavity, removing all of the intestinal contents that were 40 cm from the portion of the terminal ileum, anterior to the ileocecal junction. Before slaughter, the birds were encouraged to consume feed to avoid emptying the digestive tract, which would impair the ileal digest collection procedure.

Ileal digesta samples were lyophilized under vacuum at −40 °C for 72 h and laboratory analyses were carried out to verify the amino acid content using HPLC (high-performance liquid chromatography), according to the methodology described by the AOAC [12]. The digest dry matter and crude protein contents were also determined, according to the AOAC [12]. Insoluble acid ash and indigestibility factor were performed according to Joslyn [13]. Calculations of the standardized ileal digestibility of amino acids were performed using the methodology proposed by Sakomura and Rostagno [11], according to the equations below:CDAAapa = ((AAing − (AAdig × IF1))/AAing) × 100
CDAAsta = (AAing − ((AAdig × IF1) − (AAend × IF2))/AAing) × 100
in which CDAAapa = apparent amino acid digestibility coefficient; AAing = ingested amino acid; AAdig = digesta amino acid; IF1 = indigestibility factor 1, IF1 = AIAdiet/AIAdigesta; IF2 = indigestibility factor 2, IF2 = AIA protein-free diet/AIAdigesta; CDAAsta = standardized amino acid digestibility coefficient; and AAend = endogenous amino acid.

### 2.5. Statistical Analysis

The data collected were statistically evaluated by analysis of variance at a 5% significance level by using the ExpDes.pt package of the R statistical software (R Software v. 4.0.4). A Shapiro–Wilk test was used to determine the normality of residuals of the data; subsequently, they were subjected to analysis of variance (ANOVA). The Tukey test was used at a 5% significance level.

## 3. Results

Crude protein and amino acid analyses of the soybean meal are shown in Table 3.

Table 4 displays the characteristics of different samples of soybean meal (SBM). The samples exhibit slight variations in terms of dry matter, gross energy, and crude protein, with some samples showing higher levels of trypsin inhibitors and soluble protein. Urease activity remains consistent across the samples.

The descriptive analysis of enzyme RONOZYME^®^ ProAct recovery is shown in Table 5.

There was a significant difference (*p* < 0.05) in AME and AMEn values for the different types of soybean meal used (Table 6). Furthermore, there was a significant difference when protease was added to the broiler diet, resulting in increased AME and AMEn values of the dietary treatment (*p* < 0.05). The AME values were 2567 Kcal/kg of dry matter (DM) for the treatment without enzymatic addition (WTE) and 2677 Kcal/kg of DM for the treatment with enzymatic addition (WE). The AMEn values were 2494 Kcal/kg of DM for WTE and 2590 Kcal/kg of DM for WE.

These data indicate that enzyme inclusion increases the apparent metabolizable energy and the apparent metabolizable energy corrected for nitrogen. No interaction was observed between the factors soybean meal (SBM) and enzyme (ENZ) (ENZ * SBM) for all of the variables presented (Table 6).

No significant interaction was observed between the treatments (with and without protease supplementation) and the use of soybean meal from different regions for the apparent ileal digestibility of the amino acids (*p* > 0.05; Table 7). A significant interaction was only observed between the type of soybean meal and the enzyme for histidine (*p* < 0.05). Regarding methionine (Met), the mean values ranged from 89.03–92.35% in treatments without protease and from 90.91–94.20% with protease.

The highest values of apparent ileal digestibility were observed for lysine and arginine in both conditions (with and without protease supplementation). On the other hand, the amino acids with lower values of apparent ileal digestibility were threonine and histidine (Table 7). Overall, the results demonstrate that the apparent ileal digestibility of amino acids in soybean meal can vary according to the geographical region of production, as well as the addition of protease, which can improve its digestibility (Table 7).

Supplementation with protease increased the standardized ileal digestibility of essential amino acids compared to non-supplemented soybean meal (Table 8). Protease supplementation improved (*p* < 0.05) the standardized ileal digestibility of essential amino acids compared to non-supplemented soybean meal, although the effect of protease varied among the amino acids. The standardized ileal digestibility of essential amino acids also varied between the samples of soybean meal from different regions (Table 8). 

There was a significant interaction (*p* < 0.05) (Enz×SBM) only for the standardized digestibility of histidine (His). With protease supplementation, the average lysine digestibility increased to values between 91.70–94.18%. As for methionine digestibility, it ranged from 92.46–94.33% across different regions. With protease supplementation, the average methionine digestibility increased to values between 92.64–96.04%. In summary, protease supplementation resulted in an improvement (*p* < 0.05) in the digestibility of lysine and methionine in all analyzed regions.

## 4. Discussion

There were no significant interactions observed between soybean meals from different regions of the country and the enzyme used. Instead, only individual effects of the enzyme and the tested ingredient were identified. Soybean meal is an important protein source used in animal feed, but it contains antinutritional factors that reduce its bioavailability. The industry aims to find soybean meals with reduced antinutritional activity to improve animal performance. Commonly used ingredient composition tables such as the NRC [14], FEDNA [15], Rostagno et al. [16], and Rostagno et al. [10] generally provide nutrient profiles for soybean meal based on its crude protein (CP) content.

Although these publications provide consistent information on the amino acid (AA) profile and nitrogen (N) digestibility of soybean meal relative to CP, they do not consider factors such as soybean genotype, processing conditions, and region of origin, which can influence its chemical composition [17].

Therefore, the digestibility values varied among different producing regions, highlighting the influence of geographic location on the nutritional quality of soybean meal [18]. Based on the presented results, it can be inferred that the amino acid digestibility in soybean meals may differ due to the geographic origin of the meal. Coca-Sinova et al. [19] reported 21-day-old broilers fed with SBM from different regions, which resulted in differences in amino acid digestibility. Therefore, these authors demonstrated variation in the chemical composition and protein quality of SBM.

The difference in amino acid digestibility among different regions of soybean meal production can be influenced by factors such as variations in soil characteristics, plant genetics, agricultural practices, crop variation, fertilization, the climate of the production region, ingredient processing, and storage methods [20].

This variation in amino acid digestibility among soybean meal production regions highlights the importance of conducting detailed analyses of the nutritional composition of the meal, considering its geographical origin. This allows for a more precise formulation of diets and feeds, considering the specific characteristics of soybean meal from each region, aiming to meet the nutritional needs of animals more efficiently [21].

Soybean meal is considered highly digestible for poultry, but there are still possibilities to improve its nutritional value. Several studies have shown that the addition of proteolytic enzymes can maximize protein digestibility and metabolizable energy in high-performance broiler diets [9,10,11,12,13,14,15,16,17,18,19,20,21,22,23]. Therefore, analyzing the interaction between the origin of soybean meal and the addition of exogenous enzymes is required.

The appropriate choice of protease type and precise dosage is important to achieve the best results [24]. As demonstrated in the present research, protease was effective in improving amino acid digestibility in all soybean meals. One explanation for these results is that proteolytic enzymes can catalyze the breakdown of peptide bonds found in feed proteins [25].

Belonging to class-3 enzymes, also known as hydrolases, and the subclass called peptide hydrolases or peptidases [26], these enzymes form a broad family that can be divided into endopeptidases or proteinases and exopeptidases, depending on the position at which cleavage occurs in the peptide chain.

Whereas exopeptidases can break peptide bonds near the amino or carboxyl-terminal group of the substrate, being classified as aminos and carboxypeptidases, respectively, endopeptidases are responsible for cleaving peptide bonds further from the terminal group of the substrate [27]. The main mechanism responsible for the improvement in digestibility seems to be the increased hydrolysis of proteins in feed and the increased solubility of proteins [28] and the ability to break peptide bonds present in soybean meal proteins, facilitating the release and absorption of amino acids in the intestinal lumen and, consequently, enhancing their absorption [29].

Several authors report that the inclusion of exogenous proteolytic enzymes can offer a beneficial potential by increasing proteolytic activity in young animals, resulting in the release of smaller-sized peptides and facilitating the action of endogenous enzymes [30,31,32]. As observed in the present research, the addition of enzymes improved the digestibility of certain amino acids. Increasing the availability of amino acids is crucial, especially in the case of methionine, as it is the first limiting amino acid in diets for broilers. This amino acid has primary functions, such as being a methyl group donor with cystine and cysteine for proper feathering [33].

These results indicate that both protease supplementation and the production region of soybean meal can influence amino acid digestibility, resulting in a significant increase in nutrient utilization by animals. The supplementation of protease improves the apparent and standardized ileal digestibility of amino acids. However, when compared to the SBMxENZ interaction, histidine was the only amino acid that showed an increase in the apparent and standardized ileal digestibility coefficient.

The formulation of more effective and precise diets to non-ruminants, aiming to reduce costs and the negative impact on environmental due to the increase of nitrogen excretion, is possible by determining the digestibility coefficients of amino acids of different feed ingredients [18].

Stefanello et al. [34] conducted a study to evaluate the effects of exogenous protease added to diets containing soybean meal from two geographic regions of Brazil (south and north); they also observed interactions between soybean meal and protease, corroborating this research. According to the findings of this research, Stefanello et al. [34] report that the utilization of energy and amino acids from soybean meal depends on its origin, but supplementation with protease improves its utilization regardless of the soy source.

Additionally, the inclusion of proteolytic enzyme supplementation allows for the greater degradation of anti-nutritional factors, increasing the availability of amino acids for muscle development and formation [35]. Combining proteolytic enzyme supplementation with diets based on easily digestible amino acids can result in cost reduction and the minimization of environmental impacts. The results suggest that supplementation with protease resulted in a significant increase in AME and AMEn values and the digestibility of amino acids in soybean meals from different geographic origins in Brazil.

## 5. Conclusions

The supplementation of the enzyme RONOZYME^®^ ProAct can be an effective strategy to improve the digestibility of amino acids and increase energy utilization in diets for broilers containing soybean meal from different regions of the country.

## Figures and Tables

**Table 1 animals-14-00782-t001:** Ingredients and analyzed nutrient composition of the basal diet to broiler chickens from 14 to 28 d of age.

Ingredients	Kg/Diet
Corn	54.861
Soybean meal	36.698
Soybean oil	4.111
Dicalcium phosphate	1.656
Limestone	1.018
Common salt	0.483
Corn starch	0.300
L-lysine HCl (98%)	0.162
DL-methionine (99%)	0.277
L-Threonine	0.040
Vitamin supplement ^1^	0.110
Mineral supplement ^2^	0.110
Choline chloride (60%)	0.100
Salinomicina (12%) ^3^	0.055
Avilamicina (10%) ^4^	0.010
Antioxidant ^5^	0.010
Total	100.00
Calculated composition	
Metabolizable energy (kcal/kg)	3075
Crude protein (%)	21.252
Digestive lysine (%)	1.174
Digestible methionine + cystine (%)	0.846
Digestive threonine (%)	0.763
Tryptophan digestible (%)	0.240
Calcium (%)	0.894
Available phosphorus (%)	0.420
Sodium (%)	0.210

^1^ Composition per kg of product: Vit. A—9,000,000.00 UI; Vit. D3—2,500,000.00 UI; Vit. E—20,000.00 Mg; Vit. K3—2500.00 mg; Vit. B1—2000.00 mg; Vit. B2—6000.00 mg; Vit. B12—15.00 mg; Niacin—35,000.00 mg; Pantothenic acid—12,000.00 mg; Vit B6—8000.00 mg; Folic acid—1500.00 mg; Selenium—250.00 mg; Biotin—100.00 mg; ^2^ Composition per kg of product: Iron—100,000.00 mg; Copper—20.00 g; Manganese—130,000.00 mg; Zinc—130,000.10 mg; Iodine—2000.00 mg. ^3^ Anticoccidian (Coxistac). ^4^ Growth promoter (Surmax). ^5^ Butyl hydroxy toluene (BHT).

**Table 2 animals-14-00782-t002:** Chemical composition of protein-free diet (PFD) and experimental diets.

Ingredients	PFD (%)	PFD + SBM (%)
Corn starch	81.240	41.240
Soybean meal (1 to 9)	-	40.000
Sugar	5.000	5.000
Soybean oil	5.000	5.000
Dicalcium phosphate	2.100	2.100
Limestone	0.700	0.700
salt	0.450	0.450
Corn cob	4.000	4.000
Vitamin supplement ^1^	0.150	0.150
Mineral supplement ^2^	0.150	0.150
Choline chloride (60%)	0.200	0.200
Antioxidant ^3^	0.010	0.010
Insoluble acid ash (Celite^TM^)	1.000	1.000
Total	100.000	100.000

^1^ Composition per kg of product: vit. A. 12.000.000 IU; vit. D3. 2.200.000 IU; vit. E 30.000 IU; Vit. B1. 2.200 mg; vit B2. 6.000 mg; vit. B6. 3.300 mg; Pantothenic acid 13.000 mg; biotin. 110 mg; vit. K3. 2.500 mg; folic acid. 1.000 mg; nicotinic acid 53.0000 mg; niacin. 25.000 mg; vit. B12. 16.000 μg; selenium. 0.25 g; antioxidant 120.000 mg; and vehicle QSP. 1.000 g. ^2^ Composition per kg of product: manganese. 75.000 mg; iron. 20,000 mg; zinc. 50.000 mg; copper. 4.000 mg; cobalt. 200 mg; iodine 1.500 mg and vehicle qsp. 1.000 g. ^3^ Butyl hydroxy toluene. SMB: soybean meal.

**Table 3 animals-14-00782-t003:** Analyzed crude protein (CP) and total amino acid level (%) of different soybean meals (SBM) fed to broilers from 14 to 23 days of age.

SBM	1	2	3	4	5	6	7	8	9
CP	47.14	47.11	46.85	46.64	46.83	49.25	49.14	49.51	46.17
Lys	2.838	2.910	2.869	2.872	2.938	3.029	3.019	3.050	2.853
Thr	1.845	1.832	1.846	1.829	1.855	1.909	1.921	1.925	1.810
Met	0.615	0.621	0.614	0.616	0.629	0.640	0.643	0.645	0.617
Arg	3.483	3.459	3.441	3.425	3.491	3.619	3.63	3.655	3.395
His	1.235	1.225	1.233	1.219	1.237	1.278	1.276	1.287	1.198
Ile	2.213	2.172	2.175	2.156	2.180	2.286	2.299	2.307	2.113
Leu	3.659	3.599	3.626	3.590	3.634	3.792	3.813	3.823	3.529
Phe	2.454	2.414	2.419	2.397	2.425	2.544	2.559	2.570	2.347
Val	2.278	2.249	2.263	2.242	2.268	2.355	2.368	2.375	2.205

**Table 4 animals-14-00782-t004:** Soybean meal (SBM) characteristics.

SBM	DM (%)	GE, Kcal/Kg MN	CP (%)	TI, mg TI/g	Urease	KOH (%)	SP (%)
1	88.96	4141	47.19	1.75	0.007	81.65	38.53
2	87.89	4134	46.53	2.31	0.013	82.85	38.55
3	87.76	4151	46.87	1.34	0.002	79.60	37.31
4	88.18	4144	46.93	1.73	0.019	80.74	37.89
5	88.73	4151	47.60	2.05	0.017	83.40	39.70
6	87.98	4139	49.13	2.52	0.013	79.16	39.89
7	88.09	4154	49.28	1.79	0.007	78.19	38.53
8	88.34	4148	48.85	2.65	0.013	80.65	39.40
9	88.17	4137	45.96	1.47	0.019	82.03	37.7

DM: dry matter; GE: gross energy; CP: crude protein; TI: trypsin inhibitor; KOH: solubility in potassium hydroxide; SP: soluble protein.

**Table 5 animals-14-00782-t005:** RONOZYME^®^ ProAct recovery in experimental diets (FTT/kg).

SBM	SBM + ENZ		PFD + SBM + ENZ	
	Expected	Analyzed	Expected	Analyzed
1	15,000	20,160	15,000	17,460
2	15,000	21,790	15,000	12,970
3	15,000	18,670	15,000	15,910
4	15,000	19,850	15,000	16,940
5	15,000	19,990	15,000	17,670
6	15,000	19,270	15,000	17,390
7	15,000	17,420	15,000	18,000
8	15,000	17,520	15,000	16,720
9	15,000	16,970	15,000	17,160

SBM: soybean meal; ENZ: RONOZYME ProAct enzyme; PFD: protein-free diet.

**Table 6 animals-14-00782-t006:** Mean values of apparent metabolizable energy (AME) and apparent metabolizable energy corrected by nitrogen balance (EMAn) of soybean meal without and with the inclusion of protease, based on dry matter (DM) and natural matter (NM).

		AME. Kcal/Kg of DM	AMEn. Kcal/Kg of DM	AMEn. Kcal/Kg of NM
Enzyme	Without	2567 B	2494 B	2265 B
	With	2677 A	2590 A	2362 A
Soybean meal	1	2659 A	2592 AB	2364 A
	2	2442 B	2329 C	2146 B
	3	2540 AB	2442 BC	2229 AB
	4	2758 A	2749 A	2432 A
	5	2677 AB	2620 AB	2375 A
	6	2621 AB	2539 ABC	2305 AB
	7	2530 AB	2428 BC	2228 AB
	8	2619 AB	2467 BC	2313 AB
	9	2753 A	2713 A	2427 A
Anova	SBM	<0.0005	<0.0001	<0.0001
	Enz	0.0020	0.0056	0.0025
*p*-Value	SBM×nz	0.9966	0.9988	0.9965
	SEM	19.25	20.30	19.45

SBM: soybean meal; Enz: enzyme. Different letters in the same column indicate statistical difference. Tukey (*p* < 0.05). SEM: standard error mean.

**Table 7 animals-14-00782-t007:** Apparent ileal digestibility coefficients of essential amino acids from soybean meal samples from different regions, without or with protease supplementation.

AA	Enz	Soybean Meal										SEM	*p* Value		
		1	2	3	4	5	6	7	8	9	Mean		SBM	Enz	SBM × Enz
Lys	Without	92.8	92.0	88.9	90.8	89.7	89.7	90.7	90.4	90.8	90.7 B	0.1427	<0.001	<0.001	0.891
	With	93.0	92.6	90.4	91.5	90.7	90.3	91.3	90.7	91.8	91.4 A				
Mean		92.9 a	92.3 a	89.7 d	91.2 bc	90.2 cd	90.0 cd	91.0 bc	90.6 cd	91.3 bc					
Thr	Without	87.0	84.7	82.3	87.3	83.0	83.5	83.3	83.6	83.9	84.3 B	0.2177	<0.001	<0.001	0.227
	With	88.9	88.1	85.6	88.5	84.9	83.7	85.3	86.2	85.9	86.3 A				
Mean		87.9 a	86.4 b	83.9 c	87.9 a	83.9 c	83.6 c	84.3 c	84.9 c	84.9 c					
Met	Without	90.5	90.3	89.0	92.4	91.0	91.6	91.4	91.3	92.3	91.1 B	0.1657	<0.01	<0.01	0.140
	With	93.2	93.1	90.9	92.8	91.6	93.6	93.0	91.9	94.2	92.7 A				
Mean		91.9 a	91.8 a	90.0 c	92.6 a	91.3 bc	92.6 a	92.2 a	91.7 bc	93.3 a					
Arg	Without	93.6	92.4	91.2	93.5	93.4	93.9	94.4	93.7	94.0	93.4 B	0.1177	<0.001	<0.001	0.354
	With	94.3	94.0	92.7	94.2	94.4	94.8	94.7	93.8	95.0	94.2 A				
Mean		94.0 a	93.2 b	92.0 c	93.9 a	93.9 a	94.3 a	94.6 a	93.8 a	94.5 a					
His	Without	94.3	93.0	91.2	92.8	90.3	90.6	91.0	89.9	90.7	91.5 B	0.1543	<0.001	<0.001	0.005
	With	94.1	93.4	92.0	93.5	90.8	92.4	93.2	92.3	93.2	92.8 A				
Mean		94.2 a	93.2 b	91.7 cd	93.1 b	90.6 d	91.5 cd	92.1 c	91.0 cd	92.0 c					
Ile	Without	89.0	87.5	84.9	88.1	87.4	88.4	88.8	88.0	88.5	87.8 B	0.1557	<0.001	<0.001	0.913
	With	90.2	88.7	86.7	90.1	88.4	88.6	89.8	88.8	89.6	89.0 A				
Mean		89.6 a	88.1 a	85.8 b	89.1 a	87.9 a	88.5 a	89.3 a	88.4 a	89.0 a					
Leu	Without	90.0	88.6	86.2	89.3	87.5	88.8	88.9	87.9	88.5	88.4 B	0.1640	<0.001	<0.001	0.923
	With	91.2	90.1	88.1	90.8	88.3	89.3	90.5	89.0	90.3	89.8 A				
Mean		90.6 a	89.4 a	87.1 d	90.0 a	87.9 cd	89.0 bc	89.7 a	88.7 bc	89.4 a					
Phe	Without	90.4	89.1	87.0	89.5	88.6	89.6	89.8	89.2	89.6	89.2 B	0.1540	<0.001	<0.001	0.444
	With	91.9	91.0	89.4	91.2	89.0	89.7	90.7	89.9	90.7	90.4 A				
Mean		91.1 a	90.0 a	88.2 c	90.3 a	88.9 bc	89.7 a	90.2 a	89.6 b	90.2 a					
Val	Without	87.9	86.2	83.5	86.7	84.8	85.8	86.7	85.7	86.3	86.0 B	0.1884	<0.001	<0.001	0.926
	With	88.5	87.1	85.1	89.0	86.6	87.0	88.1	87.3	88.5	87.5 A				
Mean		88.2 a	86.7 a	84.3 c	87.9 a	85.7 b	86.5 a	87.4 a	86.5 a	87.4 a					

AA = amino acid; Lys = lysine; Thr = threonine; Met = methionine; Arg = arginine; His = histidine; Ile = isoleucine; Leu = leucine; Phe = phenylalanine; Val = valine. SBM = soybean meal; Enz = protease enzyme; SBM × Enz = interaction between soybean meal and enzyme; SEM: standard error mean. a–d: values with the same lowercase letters in the same line do not differ significantly by Tukey test (*p* < 0.05); A,B: values with the same capital letters in the same column for each analysis do not differ significantly by Tukey test (*p* < 0.05).

**Table 8 animals-14-00782-t008:** Standardized ileal digestibility coefficients of essential amino acids from soybean meal samples from different regions, without or with protease supplementation.

AA	Enz	Soybean Meal										SEM	*p* Value		
		1	2	3	4	5	6	7	8	9	Mean		SBM	Enz	SBM × Enz
Lys	Without	93.7	92.9	89.8	91.9	90.8	90.7	91.6	91.3	91.8	91.6 B	0.1414	<0.001	<0.001	0.881
	With	94.2	93.8	91.7	93.0	92.1	91.5	92.6	92.0	93.0	92.6 A				
Mean		93.9 a	93.4 a	90.8 e	92.4 bc	91.5 cde	91.0 a	92.0 bcde	91.7 cde	92.4 bcd					
Thr	Without	91.3	89.4	86.8	91.9	87.7	87.7	87.5	87.9	88.3	88.7 B	0.2141	<0.001	<0.001	0.205
	With	94.7	94.4	91.7	94.7	91.1	89.4	91.1	92.0	91.8	92.3 A				
Mean		93.0 a	91.9 a	89.2 b	93.3 a	89.4 b	88.6 b	89.8 b	90.0 b	90.0 b					
Met	Without	92.6	92.4	91.1	94.5	93.1	93.4	93.3	93.2	94.2	93.1 B	0.1592	<0.01	<0.01	0.145
	With	96.0	96.0	93.8	95.6	94.4	96.0	95.5	94.5	96.8	95.4 A				
Mean		94.3 a	94.2 a	92.5 c	95.0 a	93.8 bc	94.7 a	94.4 a	93.9 bc	95.5 a					
Arg	Without	94.6	93.5	92.2	94.6	94.6	94.8	95.4	94.9	95.1	94.4 B	0.1169	<0.001	<0.001	0.336
	With	95.7	95.5	94.2	95.7	96.0	96.2	96.0	95.2	96.5	95.7 A				
Mean		95.2 a	94.5 b	93.2 c	95.2 a	95.3 a	95.5 a	95.7 a	95.0 a	95.8 a					
His	Without	95.1	94.0	92.2	93.7	91.3	91.5	91.9	90.7	91.7	92.5 A	0.1539	<0.001	<0.001	0.006
	With	95.3	94.7	93.3	94.8	92.1	93.6	94.4	93.6	94.5	94.0 B				
Mean		95.2 a	94.4 a	92.7 bc	94.3 a	91.8 c	92.6 bc	93.1 b	92.1 bc	93.1 b					
Ile	Without	90.7	89.4	86.8	90.0	89.3	90.1	90.4	89.7	90.3	89.6 B	0.1742	<0.001	<0.001	0.891
	With	92.6	91.2	89.3	92.7	91.0	91.0	92.0	91.1	92.0	91.4 A				
Mean		91.7 a	90.3 a	88.0 b	91.3 a	90.1 a	90.5 a	91.2 a	90.4 a	91.2 a					
Leu	Without	91.5	90.3	87.9	91.0	89.2	90.3	90.4	89.5	90.1	90.0 B	0.1629	<0.001	<0.001	0.917
	With	93.3	92.4	90.3	93.1	90.7	91.4	92.6	91.8	92.5	92.0 A				
Mean		92.4 a	91.3 ab	89.1 d	92.1 a	90.0 cd	90.8 bc	91.5 a	90.6 bc	91.3 ab					
Phe	Without	92.0	90.9	88.8	91.4	90.5	91.3	91.4	90.9	91.4	91.0 B	0.1512	<0.001	<0.001	0.429
	With	94.1	93.4	91.8	93.8	91.7	92.0	92.9	92.2	93.2	92.8 A				
Mean		93.1 a	92.2 a	90.3 c	92.6 a	91.1 bc	91.6 abc	92.2 a	91.6 abc	92.3 a					
Val	Without	90.2	88.8	86.1	89.2	87.3	88.9	88.9	88.1	88.7	88.4 B	0.1890	<0.001	<0.001	0.9108
	With	91.7	90.6	88.7	92.5	90.1	90.2	91.1	90.6	91.7	90.8 A				
Mean		91.0 a	89.7 a	87.4 c	90.9 a	88.7 b	89.2 a	90.0 a	89.3 a	90.2 a					

AA = amino acid; Lys = lysine; Thr = threonine; Met = methionine; Arg = arginine; His = histidine; Ile = isoleucine; Leu = leucine; Phe = phenylalanine; Val = valine; SBM = soybean meal; Enz = protease enzyme; SBM × Enz = interaction between soybean meal and enzyme; SEM: standard error mean; a–e: values with the same lowercase letters in the same line do not differ significantly by Tukey test (*p* < 0.05); A,B: values with the same capital letters in the same column for each analysis do not differ significantly by Tukey test (*p* < 0.05).

## Data Availability

Not applicable.

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
