# Peer review of "Metabolizable Energy and Amino Acid Digestibility of Soybean Meal from Different Sources for Broiler Chickens Supplemented with Protease"

_animals, 2024, doi:10.3390/ani14050782_

Round 1
Reviewer 1 Report
Dear Authors,
There have been many previous studies on the use of enzymes and soy varieties. You need to make it clear what this study differs from the previous one.
Line 76: You need to write the light application more clearly.
Initially, 720 animals were to have been used. It was not stated how many animals were placed in groups on the 14th day of the treatment. You need to specify how many animals there are in groups.
You need to explain why the animals were placed in the cage.
Your statistical model is wrong.
Why are there no standard errors in Tables?
Table 3: Why the difference in these characteristics of the nine sources of soybeans identified in the study was not statistically determined.
Line 194: Replace “P<0.05” with “P>0.05”.
Line 195: Replace “P>0.05” with “P<0.05”.
Lines 198-201: Please check this sentence again. The indicated averages seem to be the averages of soybeans.
Table 6: Please check this sentence “a–e: Lowercase letters differ from each other on the same line”
Table 6: Please check this “SR = Santa Rosa; F. Beltrão= Francisco Beltrão; O. = Olfar;”!!!
Table 6: What does "and" mean?
Lines 226-227: “The average digestibility of lysine ranged from 89.66% to 93.66% across different regions”. This sentence is wrong. Please check again. Also, after the sentence about histidin, this sentence is not appropriate.
Lines 270-272: You stated that, "As demonstrated in the present research, the protease was effective in improving the digestibility of amino acids in soybean meals from different regions of the country.". But the enzyme-soya interaction was only effective in histidine amino acid. Accordingly, this sentence is not true. The availability of other amino acids is influenced by the main treatments.
Lines: 318-320: You need to rewrite this part. The interaction only affects the digestibility of the amino acid histidine.
Author Response
Reviewer #1 Comments:
There have been many previous studies on the use of enzymes and soy varieties. You need to make it clear what this study differs from the previous one.
Author’s Response: Thanks for the comment. This study stands out by focusing on the role of proteases in various soybean meals, providing a specialized exploration of their effects on key nutritional parameters, thereby contributing innovative insights to the existing body of research in poultry nutrition.
Line 76: You need to write the light application more clearly.
Author’s Response: The lighting program consisted of 24 lights and florescent lamps were used to provide artificial light at an intensity of 10-lux 24 h day. Thank you for your comment.
Initially, 720 animals were to have been used. It was not stated how many animals were placed in groups on the 14th day of the treatment. You need to specify how many animals there are in groups.
Author’s Response: Thanks for the comment. We apologize for the mistake. We have changed it.
You need to explain why the animals were placed in the cage.
Author’s Response: The birds were placed in metabolic cages for the collection of excreta. Thank you for your suggest.
Your statistical model is wrong.
Author’s Response: The statistical model was withdrawn. Thank you!
Why are there no standard errors in Tables?
Author’s Response: Thanks for the comment. We apologize for the mistake. We provided the standard errors.
Table 3: Why the difference in these characteristics of the nine sources of soybeans identified in the study was not statistically determined.
Author’s Response: Thank you for the comment. As these are characteristics of soybean meal, we do not think it needs to be determined statistically as this is not the intention of the study.
Line 194: Replace “P<0.05” with “P>0.05”.
Author’s Response: Thanks for the comment. We have changed it and we apologize for the mistake.
Line 195: Replace “P>0.05” with “P<0.05”.
Author’s Response: Thanks for the comment. We have changed it and we apologize for the mistake.
Lines 198-201: Please check this sentence again. The indicated averages seem to be the averages of soybeans.
Author’s Response: Thank you for your comment. We checked and corrected it.
Table 6: Please check this sentence “a–e: Lowercase letters differ from each other on the same line”
Author’s Response: Thanks for the comment. We have changed it and we apologize for the mistake.
Table 6: Please check this “SR = Santa Rosa; F. Beltrão= Francisco Beltrão; O. = Olfar;”!!!
Author’s Response: Thanks for the comment. We have changed it and we apologize for the mistake.
Table 6: What does "and" mean?
Author’s Response: This was wrong. We apologize for that. We changed it for ‘or’.
Lines 226-227: “The average digestibility of lysine ranged from 89.66% to 93.66% across different regions”. This sentence is wrong. Please check again. Also, after the sentence about histidin, this sentence is not appropriate.
Author’s Response: Thanks for the comment. We have changed it and we apologize for the mistake.
Lines 270-272: You stated that, "As demonstrated in the present research, the protease was effective in improving the digestibility of amino acids in soybean meals from different regions of the country.". But the enzyme-soya interaction was only effective in histidine amino acid. Accordingly, this sentence is not true. The availability of other amino acids is influenced by the main treatments.
Author’s Response: Thank you for your comment. We tried to showed an individual effect of protease and soybean meal. As suggested by you, we rewrote this part and tried to demonstrate it more clearly.
Lines: 318-320: You need to rewrite this part. The interaction only affects the digestibility of the amino acid histidine.
Author’s Response: Thank you for your comment. There was an individual effect of the region of soybean meal production and an individual effect of the enzyme. The interaction of soybean meal and enzyme showed an effect on histidine digestibility coefficient. We rewritten it to be more understandable.

Reviewer 2 Report
Additional comments: The experimental design needs to be re-examined. The authors mentioned that a 9 x 2 + 1 design was used. However, most of the tables presented showed a 9 x 2 factorial design.
The quality of the English Language is okay.
Author Response
Reviewer #2 Comments:
Additional comments: The experimental design needs to be re-examined. The authors mentioned that a 9 x 2 + 1 design was used. However, most of the tables presented showed a 9 x 2 factorial design.
Author’s Response: Thank you for the comment. The reference diet is used for digestibility calculations and it was a reference for the formulation of experimental diets, changing only the source of soybean meal.

Reviewer 3 Report
This manuscript aims to investigate the effect of the serine protease on the amino acid’s digestibility and metabolizable energy of soybean meal from different locations for broilers. There are some major questions/concerns that need to be addressed.
Major Comments:
1. The description of experimental design was not clearly, please revise.
2. I suggest the author provide data of growth performance, CP metabolizable rate to further illustrate the effects between different treatments on broilers.
3. The amino acid profile of the nine soybean meal should be provided.
4. Standard error of mean should be used in instead of CV.
5. In DISCUSSION is too general, there was no discussion on different sources of soybean meal and the reasons for the differences in nitrogen metabolism.
Specific Comments:
L13: “the value”was mistyped twice.
L99: should be “from d14 to d23”.
Table 2: Full names of “Suppl. Vitam and Suppl. Min.” were missed.
Author Response
Reviewer #3 Comments:
- The description of experimental design was not clearly, please revise.
Author’s Response: Thank you for the comment. We revised it.
- I suggest the author provide data of growth performance, CP metabolizable rate to further illustrate the effects between different treatments on broilers.
Author’s Response: The objective of the research was to evaluate the digestibility of nutrients and energy values so it does not have the performance.
- The amino acid profile of the nine soybean meal should be provided.
Author’s Response: The amino acid profile of soybean meals was provided. Thank you for your comment.
- Standard error of mean should be used in instead of CV.
Author’s Response: Thank you for your comment. The SEM was provided.
- In DISCUSSION is too general, there was no discussion on different sources of soybean meal and the reasons for the differences in nitrogen metabolism.
Author’s Response: Thank you for the comment. In the penultimate paragraph, we try to discuss the influence of soybean meal from different regions.
Specific Comments:
L13: “the value”was mistyped twice.
Author’s Response: Thanks for the comment. We have changed it and we apologize for the mistake.
L99: should be “from d14 to d23”.
Author’s Response: Thanks for the comment. We have changed it.
Table 2: Full names of “Suppl. Vitam and Suppl. Min.” were missed.
Author’s Response: Thanks for the comment. We have changed it and we apologize for the mistake.

Reviewer 4 Report
Overall, the manuscript titled ‘Metabolizable energy and amino acid digestibility of soybean meal from different sources for broiler chickens supplemented with protease’ is well written and doesn't have too many shortcomings.
The only issue that reduces its novelty is the fact that the commercial enzyme (RONOZYME® ProAct), available, for example, in Europe. However, the data obtained data in the assessment of the effectiveness of increasing amino acid digestibility of amino acids and improving the energy value of various variants of soybean meal have solid scientific and practical value.
Abstract:
The abstract lacks more information on the number of birds used in the experiment (720 birds), the number of groups, the number of repetitions within a group, and the number of birds per repetition. Please, provide this information.
Introduction:
I have no major comments here. This section is very well done, clearly, with all the information related to the research topic and justifying the purpose of the study. Likewise, the authors did a very good job of the discussion section.
Materials and Methods:
The overall methodology part for broiler management, diets, feed analysis, including amino acid composition, diets, use of a digestibility marker is presented correctly (AOAC protocols, in general).
However, the experimental part on chickens requires improvements, because the authors did not indicate the exact number of birds and the number of birds per repetition. The information provided in L: 80-85 shows that the target number of birds used for the digestibility experiment was 684 and not 720 as at the beginning of the trial (‘19 treatments and six replicates of six birds per experimental unit’). If this is correct, these data should be included both in the Material and Methods section and in the Abstract to ensure transparency of the experiment design.
Ultimately, birds were transferred to wire floor cages for the duration of the experiment which lasted from the 14th to the 28th day of bird life; therefore, the final number of animals should be indicated exactly in this period. In turn, it is good that the authors included the appropriate ‘ethical statement’ in the manuscript (L: 327-331).
L: 138-143: Indicate the number of birds killed in this procedure or whether all birds from each replication were killed or only some of them?
Results:
L: 194-195: Correct the P value because it is given incorrectly for the interaction (as P>0.05); in Table 6, this value is 0.005 (so it can be presented as P<0.05, just as in L: 225, related to Table 7).
Author Response
Reviewer #4 Comments:
Overall, the manuscript titled ‘Metabolizable energy and amino acid digestibility of soybean meal from different sources for broiler chickens supplemented with protease’ is well written and doesn't have too many shortcomings.
Author’s Response: Thanks for the comment.
The only issue that reduces its novelty is the fact that the commercial enzyme (RONOZYME® ProAct), available, for example, in Europe. However, the data obtained data in the assessment of the effectiveness of increasing amino acid digestibility of amino acids and improving the energy value of various variants of soybean meal have solid scientific and practical value.
Author’s Response: Thanks for the comment.
Abstract:
The abstract lacks more information on the number of birds used in the experiment (720 birds), the number of groups, the number of repetitions within a group, and the number of birds per repetition. Please, provide this information.
Author’s Response: Thanks for the comment. We have changed it, thank you!
Introduction:
I have no major comments here. This section is very well done, clearly, with all the information related to the research topic and justifying the purpose of the study. Likewise, the authors did a very good job of the discussion section.
Author’s Response: Thanks for the comment.
Materials and Methods:
The overall methodology part for broiler management, diets, feed analysis, including amino acid composition, diets, use of a digestibility marker is presented correctly (AOAC protocols, in general).
Author’s Response: Thanks for the comment.
However, the experimental part on chickens requires improvements, because the authors did not indicate the exact number of birds and the number of birds per repetition. The information provided in L: 80-85 shows that the target number of birds used for the digestibility experiment was 684 and not 720 as at the beginning of the trial (‘19 treatments and six replicates of six birds per experimental unit’). If this is correct, these data should be included both in the Material and Methods section and in the Abstract to ensure transparency of the experiment design.
Author’s Response: Thanks for the comment. We have changed it and we apologize for the mistake.
Ultimately, birds were transferred to wire floor cages for the duration of the experiment which lasted from the 14th to the 28th day of bird life; therefore, the final number of animals should be indicated exactly in this period. In turn, it is good that the authors included the appropriate ‘ethical statement’ in the manuscript (L: 327-331).
Author’s Response: Thanks for the comment. We have changed it and we apologize for the mistake.
L: 138-143: Indicate the number of birds killed in this procedure or whether all birds from each replication were killed or only some of them?
Author’s Response: Thanks for the comment. We have changed it and we apologize for the mistake.
Results:
L: 194-195: Correct the P value because it is given incorrectly for the interaction (as P>0.05); in Table 6, this value is 0.005 (so it can be presented as P<0.05, just as in L: 225, related to Table 7).
Author’s Response: Thanks for the comment. We have changed it and we apologize for the mistake.

Round 2
Reviewer 1 Report
Dear Authors,
The author has corrected most of the mentioned parts. However, it did not adequately state the criteria according to which it determined the soybean raw materials in 9 different regions used for the research. Additionally, there is no statistical analysis of protein values related to soybean raw material. Therefore, it cannot be determined which feature of soybean obtained from these different regions will add value to the study. I think that the study in this form does not differ significantly from previous studies.
Best regards,